# Evolution of extreme resistance to ionizing radiation via genetic adaptation of DNA repair

Rose T Byrne[1†], Audrey J Klingele[1†], Eric L Cabot[2], Wendy S Schackwitz[3], Jeffrey A Martin[3], Joel Martin[3], Zhong Wang[3], Elizabeth A Wood[1], Christa Pennacchio[3], Len A Pennacchio[3], Nicole T Perna[2,4], John R Battista[5], Michael M Cox[1]*

[1]Department of Biochemistry, University of Wisconsin-Madison, Madison, United States; [2]Genome Center, University of Wisconsin-Madison, Madison, United States; [3]DOE Joint Genome Institute, Lawrence Berkeley National Laboratory, Walnut Creek, United States; [4]Laboratory of Genetics, University of Wisconsin-Madison, Madison, United States; [5]Department of Biological Sciences, Louisiana State University and A & M College, Baton Rouge, United States

**Abstract** By directed evolution in the laboratory, we previously generated populations of *Escherichia coli* that exhibit a complex new phenotype, extreme resistance to ionizing radiation (IR). The molecular basis of this extremophile phenotype, involving strain isolates with a 3-4 order of magnitude increase in IR resistance at 3000 Gy, is now addressed. Of 69 mutations identified in one of our most highly adapted isolates, functional experiments demonstrate that the IR resistance phenotype is almost entirely accounted for by only three of these nucleotide changes, in the DNA metabolism genes *recA*, *dnaB*, and *yfjK*. Four additional genetic changes make small but measurable contributions. Whereas multiple contributions to IR resistance are evident in this study, our results highlight a particular adaptation mechanism not adequately considered in studies to date: Genetic innovations involving pre-existing DNA repair functions can play a predominant role in the acquisition of an IR resistance phenotype.

*For correspondence: cox@ biochem.wisc.edu

†These authors contributed equally to this work

Competing interests: The authors declare that no competing interests exist.

## Introduction

Ionizing radiation (IR) is encountered humans in the form of medical X-rays and tumor irradiation, and very rarely in the context of nuclear power plant malfunction. The study of organisms with extreme resistance to IR has received increasing attention as a potential source of mechanistic insights that might permit the modulation of IR resistance in cells. *Deinococcus radiodurans*, which can absorb IR doses in excess of 5 kGy without lethality (over 1000 times the lethal dose for humans), has become a key model organism for understanding this phenotype (*Cox and Battista, 2005*, *Blasius et al., 2008*). Mainly through the formation of reactive oxygen species (ROS), IR can lead to the damage of protein, DNA, and all other cellular macromolecules (*Sonntag Cv, 2005*). Ongoing research into the molecular basis of extreme IR resistance has suggested three potential classes of mechanisms. These are (A) a condensed nucleoid structure, which could potentially facilitate DNA repair processes (*Zimmerman and Battista, 2005*, *Levin-Zaidman et al., 2003*), (B) an enhanced capacity for amelioration of protein damage, which could protect DNA repair systems and make them more readily available following irradiation (*Daly, 2012*, *Daly, 2009*), and (C) potential specialized pathways for DNA repair (*Zahradka et al., 2006*, *Cox and Battista, 2005*).

**eLife digest** X-rays and other forms of ionizing radiation can damage DNA and proteins inside cells. The radiation interacts with aqueous solutions to produce reactive forms of oxygen, which then cause the damage. A range of mechanisms exist to moderate and/or repair this damage, with certain species being able to tolerate extraordinary levels of radiation. The bacterium *D. radiodurans*, for example, can survive radiation levels that are over 1000 times higher than the levels that can kill human cells.

The molecular basis of high-level resistance to ionizing radiation is not well understood, and several mechanisms have been proposed. Recent work has focused on passive mechanisms that are based on changes in cellular levels of certain small molecules that prevent damage by reactive forms of oxygen molecules.

Now, based on experiments on *E. coli*, Byrne et al. demonstrate that active mechanisms, involving adaptations in the cellular DNA repair systems, can bring about dramatic increases in radiation resistance. The experiments were performed on populations of *E. coli* cells that had been subjected to an evolutionary selection for extremely high resistance to ionizing radiation. This involved exposing the *E. coli* cells to ionizing radiation that killed most of the population, and then growing up the survivors. Many repetitions of this process led to a population of cells with a resistance that was comparable to that of the bacterium *D. radiodurans.* The same evolution experiment was carried out four times, generating four separate populations of bacteria that were resistant to ionizing radiation.

Byrne et al. sequenced the genomes of the *E. coli* after 20, 40 or 50 rounds of the selection process, and compared mutations found in the four separate evolved populations. This showed that nine genes were particularly prone to mutations. Together, these genes had roles in repairing and copying DNA sequences, in decreasing damage caused by reactive forms of oxygen, and in manufacturing the molecular wall that shields cells.

To assess the importance of the mutations in the nine genes, Byrne et al. took Founder cells from the initial population of *E. coli* cells–which were not resistant to ionizing radiation–and introduced the very same mutations, one at a time. Then the mutations that had the largest positive effects on resistance to ionizing radiation were combined. Introducing particular mutations into three DNA repair genes resulted in the highest aggregate levels of resistance. Finally, evolved *E. coli* cells that were already resistant were made more sensitive to radiation by repairing the same individual mutations. Again, the biggest change was observed with the DNA repair genes. Indeed, repairing the mutations in just the three DNA repair genes completely removed the radiation resistance.

The next step is to determine how the properties of the mutated proteins change, and how those changes lead to radiation resistance. Also, there are clues in the work that suggest the presence of additional ways for cells to become radiation resistant, and these remain to be explored.

The amelioration of protein oxidation has received extensive experimental support as a mechanism to account for much if not all of the observed IR resistance in *D. radiodurans* (*Daly et al., 2004*, *Slade and Radman, 2011*, *Krisko and Radman, 2010*, *Daly, 2012*, *Daly, 2009*). Increases in cytosolic anti-oxidant capacity, particularly an increase in the cellular Mn/Fe ratio (*Daly et al., 2004*), appear to make the major contributions. Increasing the Mn/Fe ratio limits the Fenton chemistry that produces ROS (*Krisko and Radman, 2010*, *Daly, 2012*). The hypothesis that the condensed nucleoid of *Deinococcus* facilitates efficient DNA repair has been questioned, since the presence of condensed nucleoids does not correlate reliably with radiation resistance in bacterial species (*Zimmerman and Battista, 2005*).

Repair of IR-induced DNA damage is clearly important to survival. However, the constellation of DNA repair functions in *D. radiodurans* is unremarkable by bacterial standards. *D. radiodurans* and other IR resistant species appear to have approximately the same toolbox of DNA repair pathways as non-resistant species. Thus, the argument has been advanced that specialized DNA repair plays little or no role in IR resistance (*Daly, 2012*, *2009*). Instead, antioxidants prevent protein oxidation and render the classical DNA repair systems more readily available to correct the effects of IR.

Bacteria are being used in long-term laboratory experiments that have elucidated many aspects of evolutionary biology (*Barrick et al., 2009*, *Hindre et al., 2012*, *Elena and Lenski, 2003*). Directed bacterial evolution can be used as a tool to determine the molecular underpinnings of adaptation to a stress or a new environment. Does IR resistance always arise via enhanced protection of otherwise commonplace DNA repair proteins from oxidative inactivation? Alternatively, can a special facility for DNA repair or some other mechanism make a significant contribution to this extremophile phenotype?

In an earlier study, we demonstrated that the naturally IR sensitive bacterium, *E. coli,* could acquire an extreme IR resistance phenotype by directed evolution (*Harris et al., 2009*). In brief, we carried out 20 rounds of selection, with each round consisting of IR exposure sufficient to kill ~99% of the population, followed by outgrowth. The experiment was carried out four times, generating four separately evolved populations (IR-1-20, IR-2-20, IR-3-20, and IR-4-20). In the evolved populations, survival at 3000 Gy was typically increased by 3–4 orders of magnitude (*Harris et al., 2009*). We reported genomic sequences from seven isolates of IR-1-20, one from IR-2-20, and one from IR-3-20. Genetic alterations in the evolved strains were quite numerous (within the range of 40–80 per isolate), suggesting a complex and somewhat variable pattern of genetic innovation. None of the isolates in that study had mutator phenotypes relative to the Founder strain, although it is possible that mutators appear and thrive in the various populations for periods during the many cycles of selection (*Harris et al., 2009*). An approximately fivefold variation in levels of resistance to IR was evident in a screen of 62 separate isolates from population IR-1-20 (*Harris et al., 2009*).

It remained for us to determine which of those mutations underlay the increase in radiation resistance, and what each of them might contribute. If amelioration of protein oxidation uniquely explains high level IR resistance in bacteria, then the mutations that contribute substantially to the phenotype might follow a fairly predictable pattern. We now provide a greatly expanded analysis involving many additional evolved isolates. The results reveal multiple contributions to IR resistance as well as the detailed molecular basis of the phenotype in one isolate.

## Results

### Sequencing of multiple isolates from evolved populations reveals different evolutionary paths to IR resistance

In the present study, we augmented the earlier data with the complete genomic sequences of 13 new isolates from populations IR-2-20, IR-3-20, and IR-4-20. In addition, we examined the complete genomic sequences of nine isolates derived from either 20 or 30 rounds of further evolution of CB1000, an isolate derived from IR-1-20. We note that IR resistance in this more highly evolved population (IR-CB1000-30) was impossible to measure accurately. The cells in this population grew more slowly than the Founder, but growth continued during irradiation at 6 Gy/min (John R Battista, unpublished data). The results from the previous (*Harris et al., 2009*) and new sequencing efforts are combined in *Supplementary file 1A,B*. The overall selection scheme is illustrated in *Figure 1*. The results give us a much-enhanced view of mutational patterns that are likely to contribute to the acquired IR resistance.

A summary of mutation types detected in the various sequenced isolates is presented in *Table 1*. In each case, the genomic sequence was compared to the 4,639,675-bp reference genome. In the isolates derived from the original four evolved populations, there were between 44 and 77 genetic alterations. The numbers of mutations jumped appreciably in the isolates derived from the further evolution of CB1000, with 242–267 mutations present in these strains. Transition mutations dominated the mutational spectrum.

One straightforward pattern noted in the earlier study was continued. The only genetic change that is universal to all strains sequenced is deletion of the e14 prophage, a defective lambdoid prophage that is 15.4 kb in length. The deletion of occurs early, within the first three rounds of selection for IR resistance (EA Wood and JR Battista, unpublished data), and it makes a significant contribution (an approximately 10-fold increase in survival at 3000 Gy) to the overall IR resistance phenotype (*Harris et al., 2009*).

Population IR-1-20 has a particularly complex population structure and is strikingly different from the IR-2-20 and IR-3-20 populations. Outside of the e14 deletion, there are no mutations shared by all seven of the IR-1-20 isolates. There are three groupings of mutations in subsets of the clonal isolates that reflect clonal interference (*Gerrish and Lenski, 1998*, *Perron et al., 2012*,*Rozen et al., 2002*) in this population (JR Battista, unpublished results). In contrast, isolates from the IR-2-20 and IR-3-20

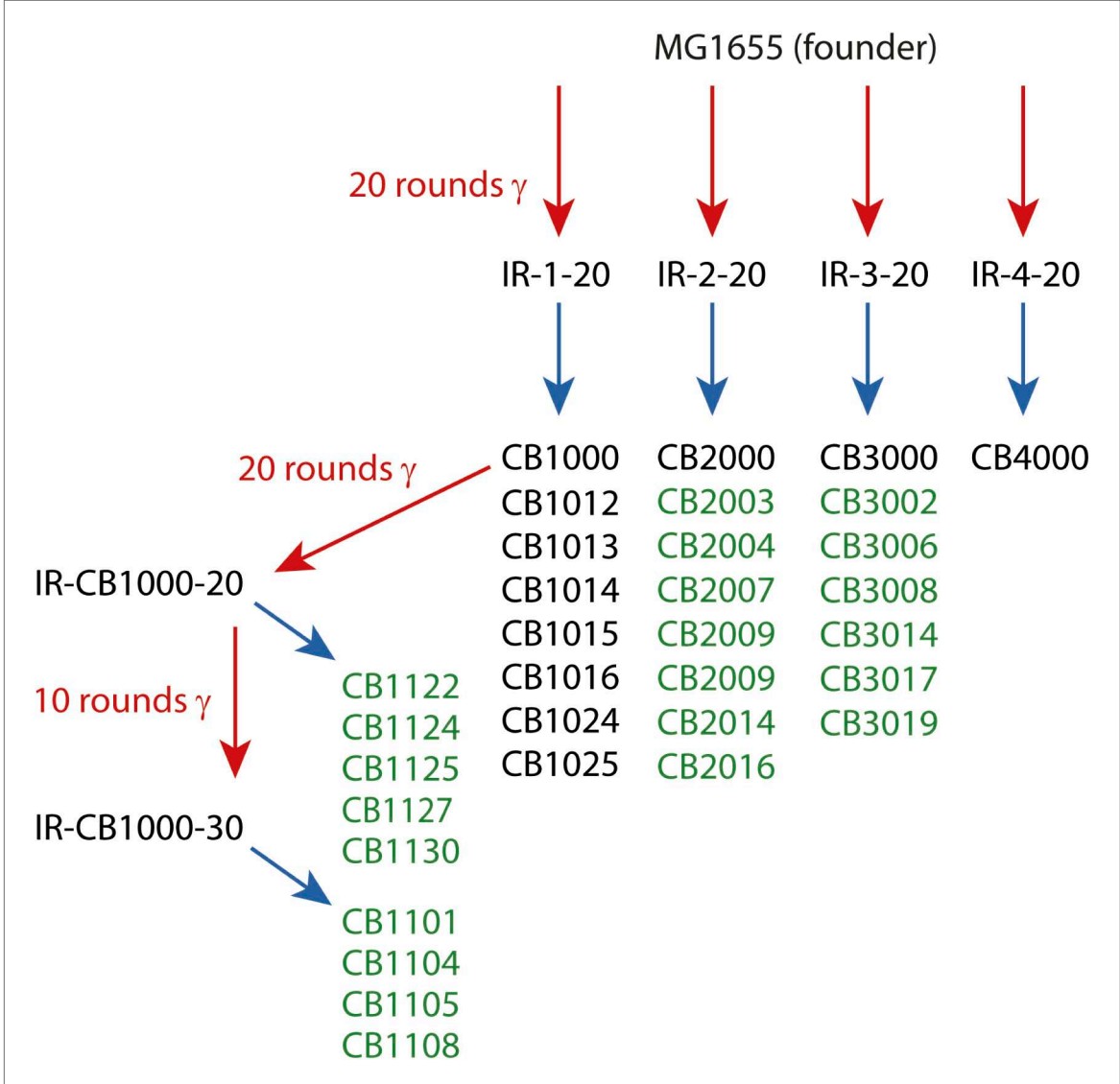

**Figure 1**. Directed evolution scheme for the evolved isolates described in this paper. Red arrows denote cycles of irradiation and outgrowth. Evolved populations have titles beginning with "IR". Isolates were derived from each listed population, as indicated by blue arrows. Isolates from each population are listed under the respective blue arrows, and each isolate features a name beginning with CB. Isolates listed in green text are described for the first time in this study. The sequences of the remaining isolates were described previously (*Harris et al., 2009*), and are listed here since the genomic data was utilized in the current analysis.

populations each exhibit a number of mutations (23 in IR-2-20 and 30 in IR-3-20) that are present in all seven isolates from their respective populations, and are effectively fixed. The apparent fixation of a mutation in one of these populations may reflect its importance to the phenotype. Alternatively, this may reflect a genetic bottleneck at some stage in the evolution of these populations, and speaks to the close relationship of the isolates.

## Identification of mutations contributing to IR resistance in IR-2-20

To identify the mutations that are most likely to contribute to the IR resistance phenotype, we focused on genetic alterations that exhibited the following criteria: (I) the mutation is present in all or most isolates sequenced in at least one population, (II) the mutated gene was a prominent mutational target in at least one other sequenced population or mutations are found in genes in the same operon or pathway in other populations. Application of the second criterion requires data that was unavailable in our earlier study (*Harris et al., 2009*), and provides a pattern implicating a particular mutation in the

**Table 1.** Summary of the mutational spectrum observed in strains derived from directed evolution of resistance to ionizing radiation

| | IR-1-20 | | | | | | | IR-2-20 | | | | | | | IR-3-20 | | | | | | | IR-4-20 | IR-CB1000-20 | | | | | IR-CB1000-30 | | | |
|---|---|---|---|---|---|---|---|---|---|---|---|---|---|---|---|---|---|---|---|---|---|---|---|---|---|---|---|---|---|---|---|
| | CB1000 | CB1012 | CB1013 | CB1014 | CB1015 | CB1024 | CB1025 | CB2000 | CB2003 | CB2004 | CB2007 | CB2009 | CB2004 | CB2016 | CB3010 | CB3002 | CB3006 | CB3008 | CB3004 | CB3017 | CB3019 | CB4010 | CB1122 | CB1124 | CB1125 | CB1127 | CB1130 | CB1101 | CB1104 | CB1105 | CB1108 |
| **Transitions** | | | | | | | | | | | | | | | | | | | | | | | | | | | | | | | |
| C → T | 19 | 20 | 6 | 10 | 15 | 6 | 9 | 12 | 13 | 11 | 12 | 11 | 9 | 9 | 14 | 14 | 13 | 14 | 13 | 12 | 12 | 14 | 59 | 64 | 69 | 69 | 63 | 60 | 62 | 63 | 62 |
| G → A | 18 | 9 | 11 | 16 | 11 | 11 | 12 | 17 | 17 | 16 | 15 | 13 | 13 | 11 | 17 | 11 | 13 | 13 | 12 | 8 | 15 | 10 | 60 | 65 | 78 | 71 | 60 | 57 | 60 | 60 | 60 |
| T → C | 7 | 7 | 6 | 4 | 6 | 13 | 4 | 11 | 8 | 16 | 10 | 9 | 9 | 11 | 11 | 4 | 13 | 11 | 7 | 6 | 9 | 7 | 51 | 51 | 40 | 44 | 47 | 47 | 47 | 47 | 47 |
| A → G | 15 | 16 | 7 | 8 | 8 | 14 | 11 | 10 | 6 | 6 | 7 | 7 | 6 | 5 | 6 | 14 | 10 | 10 | 9 | 11 | 6 | 8 | 42 | 43 | 42 | 54 | 50 | 46 | 50 | 50 | 50 |
| **Transversions** | | | | | | | | | | | | | | | | | | | | | | | | | | | | | | | |
| G → T | 4 | 4 | 1 | 1 | 1 | 3 | 5 | 3 | 4 | 2 | 3 | 5 | 4 | 2 | 5 | 5 | 4 | 4 | 4 | 6 | 4 | 2 | 10 | 9 | 7 | 8 | 6 | 5 | 6 | 6 | 6 |
| C → A | 1 | 1 | 2 | 1 | 1 | 5 | 6 | 5 | 3 | 6 | 3 | 3 | 4 | 6 | 2 | 4 | 2 | 1 | 3 | 4 | 1 | 6 | 3 | 2 | 5 | 5 | 6 | 4 | 5 | 5 | 4 |
| A → C | 1 | 1 | 2 | | | | | | 1 | | | 1 | 1 | 2 | | 1 | | 1 | 2 | 2 | 1 | 1 | 7 | 4 | 2 | 2 | 4 | 4 | 3 | 3 | 3 |
| T → G | | 1 | 1 | 1 | 1 | 1 | | | | | 1 | | | | 1 | 1 | 1 | 1 | 2 | 1 | 2 | 2 | 2 | 4 | | | | 1 | 1 | 1 | 1 |
| G → C | 1 | 1 | 1 | 2 | 2 | | | | 1 | 1 | | | 1 | | | 1 | | | 1 | | 1 | 1 | 2 | 1 | 1 | 2 | 4 | 3 | 3 | 3 | 3 |
| C → G | | 1 | | | | | | 2 | 3 | 1 | 1 | 1 | 1 | 1 | | 1 | | | 1 | | | 2 | 1 | 2 | 1 | 1 | 2 | 2 | 2 | 2 | 2 |
| A → T | | 1 | | 2 | | | | | 1 | 1 | 3 | 1 | 2 | 1 | 4 | 3 | 1 | 1 | 2 | 1 | 1 | 1 | 2 | 2 | 1 | 2 | 3 | 3 | 3 | 3 | 3 |
| T → A | | 1 | | 1 | | | | | 1 | 1 | 1 | 1 | | | 1 | 1 | 1 | 3 | 1 | 1 | 1 | 2 | 3 | 3 | 2 | 3 | 3 | 3 | 3 | 3 | 2 |
| **Insertions** | | | | | | | | | | | | | | | | | | | | | | | | | | | | | | | |
| 767bp IS1 | 2 | 1 | 1 | 1 | 1 | 1 | 1 | | | | | | | | | | | | | | | | | | | | | | | | |
| **Deletions** | 1 | | | | | | | | | | | | | | | | | | | | | | | | | | | | | | |
| e14 | 1 | 1 | 1 | 1 | 1 | 1 | 1 | 1 | 1 | 1 | 1 | 1 | 1 | 1 | 1 | 1 | 1 | 1 | 1 | 1 | 1 | 1 | 1 | 1 | 1 | 1 | 1 | 1 | 1 | 1 | 1 |
| 93bp | | | | | | | | | | | | | | | | | 1 | | | | | | | | | | | | | | |
| **Totals** | 70 | 63 | 40 | 46 | 49 | 56 | 53 | 66 | 55 | 64 | 56 | 53 | 50 | 51 | 63 | 57 | 63 | 61 | 55 | 53 | 54 | 53 | 243 | 251 | 249 | 265 | 250 | 236 | 246 | 247 | 244 |

The mutational spectrum of the population is inferred based on the genetic alterations observed in the subset of strains sequenced.

acquisition of an extreme IR resistance phenotype. This criterion also assumes that there is a significant level of phenotypic parallelism (*Hindre et al., 2012*, *Futuyama, 1986*) between the independently evolved populations. *Table 2* summarizes the results of applying these criteria, with nine altered genes or systems meeting these two criteria. The entire complement of mutations found in all isolates from the four evolved populations, both those sequenced for this study and those analyzed in the earlier study (*Harris et al., 2009*), are provided in *Supplementary file 1A*.

The prominent genetic alterations present in the various evolved isolates are not identical. We thus set out to define the genetic basis of the observed IR resistance in one representative isolate. We focused on the isolated strain CB2000 from the IR-2-20 population for several reasons. First, CB2000 is one of the most radiation resistant of the isolated strains from the original evolved populations (*Harris et al., 2009*). Second, it is particularly well characterized (*Harris et al., 2009*; and this study). Third, it was isolated from a population that lacks any sign of clonal interference, allowing us to focus on patterns present in a relatively simple population. Finally, the patterns of mutations we highlight below overlap patterns that are evident in other populations, making CB2000 a good barometer of mechanisms by which radiation resistance evolved in many of our strains. Of the 9 genes and/or systems reflected in *Table 2*, seven are represented in CB2000.

The candidate CB2000 genes can be grouped into three functional categories: (I) DNA repair and replication (*recA*, *dnaB*, *yfjK*), (II) oxidative damage suppression (*rsxB* and *gsiB*), and (III) cell wall biogenesis (*wcaK* and *nanE*). All of these represent prominent mutational patterns.

**Table 2.** Summary of prominent mutational patterns observed in multiple evolved populations

Column groups: IR-1-20 (CB1000, CB1012, CB1013, CB1014, CB1015, CB1024, CB1025); IR-2-20 (CB2000, CB2003, CB2004, CB2007, CB2019, CB2014, CB2006); IR-3-20 (CB3000, CB3002, CB3006, CB3008, CB3014, CB3017, CB3019); IR-4-20 (CB4000); IR CB1000−+

| Gene | Position | Ref. Allele | CB1000 | CB1012 | CB1013 | CB1014 | CB1015 | CB1024 | CB1025 | CB2000 | CB2003 | CB2004 | CB2007 | CB2019 | CB2014 | CB2006 | CB3000 | CB3002 | CB3006 | CB3008 | CB3014 | CB3017 | CB3019 | CB4000 | IR CB1000−+ | Change | Mutation Type |
|---|---|---|---|---|---|---|---|---|---|---|---|---|---|---|---|---|---|---|---|---|---|---|---|---|---|---|---|
| clpP | 456127 | A | G | G |  |  |  |  |  |  |  |  |  |  |  |  |  |  |  |  |  |  |  |  | + | Y75C | N |
| clpP/clpX | 456637 | G |  |  |  |  |  |  |  |  |  |  |  |  |  |  | A | A | A | A | A | A | A |  |  | - | I |
| clpX | 457803 | A |  |  |  | G |  |  |  |  |  |  |  |  |  |  |  |  |  |  |  |  |  |  |  | Y384C | N |
| gsiB | 868947 | A |  |  |  |  |  |  |  | G | G | G | G | G | G | G |  |  |  |  |  |  |  |  |  | N104S | N |
| gsiB | 869499 | T |  |  |  |  |  |  |  |  |  |  |  |  |  |  | C | C | C | C | C | C | C |  |  | L288P | N |
| gsiB | 870075 | T |  |  |  |  |  |  |  |  |  |  |  |  |  |  |  |  |  |  |  |  |  | C |  | V480A | N |
| fnr | 1396995 | A |  |  |  |  |  |  |  |  |  |  |  |  |  |  | T | T | T | T | T | T | T |  | + | F185I | N |
| rsxB | 1704735 | A |  |  |  |  |  |  |  | G | G | G | G | G | G | G |  |  |  |  |  |  |  |  | + | K121E | N |
| rsxD | 1707299 | T |  |  |  |  |  |  |  |  |  |  |  |  |  |  | C | C | C | C | C | C | C |  |  | V44A | N |
| wcaM | 2113451 | T |  |  |  |  |  |  |  |  |  |  |  |  |  |  |  |  |  |  |  |  |  | C | + | N156S | N |
| wcaK | 2116031 | T |  |  |  |  |  |  |  | C | C |  | C | C | C |  |  |  |  |  |  |  |  |  |  | Y132C | N |
| wcaC | 2129153 | A |  |  |  |  |  |  |  |  |  |  |  |  |  |  | G | G | G | G | G | G | G |  |  | S313S | S |
| yfjK | 2759609 | G |  |  |  |  |  |  |  |  |  |  |  |  |  |  |  |  |  |  |  |  |  | A |  | H651Y | N |
| yfjK | 2760683 | T |  |  |  |  |  | C |  |  |  |  |  |  |  |  |  |  |  |  |  |  |  |  |  | K293E | N |
| yfjK | 2760809 | G |  |  |  |  |  |  |  |  |  |  |  |  | A |  |  |  |  |  |  |  |  |  |  | P251S | N |
| yfjK | 2761108 | G |  |  |  |  |  |  |  | T | T | T | T | T | T | T |  |  |  |  |  |  |  |  |  | A151D | N |
| recA | 2820924 | C |  |  |  |  |  | A |  |  |  |  |  |  |  |  |  |  |  |  |  |  |  |  |  | A289S | N |
| recA | 2820962 | T |  |  | G | G | G |  | G |  |  |  |  |  |  |  |  |  |  |  |  |  |  |  |  | D276A | N |
| recA | 2820963 | C |  |  |  |  |  |  |  | T | T | T | T | T | T | T |  |  |  |  |  |  |  | T |  | D276N | N |
| nanE | 3368674 | C |  |  |  |  |  |  |  | T | T | T | T | T | T | T |  |  |  |  |  |  |  |  | + | A128T | N |
| nanT | 3369380 | A |  |  |  |  |  |  |  |  |  |  |  |  |  |  | G | G | G | G | G | G | G |  |  | F405S | N |
| dnaB | 4262560 | T |  |  |  |  |  | C |  |  |  |  |  |  |  |  |  |  |  |  |  |  |  |  | + | L74S | N |
| dnaB | 4262578 | C |  |  |  |  |  |  |  | A | A | A | A | A | A | A |  |  |  |  |  |  |  |  |  | P80H | N |
| dnaB | 4262935 | C |  |  |  |  |  |  |  |  |  |  |  |  |  |  |  |  |  |  |  |  |  | A |  | P199Q | N |
| priA | 4123174 | C |  |  |  |  |  |  |  |  |  |  |  |  |  |  | T | T | T | T | T | T | T |  |  | V553I | N |
| priC | 489549 | A | G | G |  |  |  |  |  |  |  |  |  |  |  |  |  |  |  |  |  |  |  |  |  | L162P | N |
| dnaT | 4599105 | G |  |  | A | A | A |  | A |  |  |  |  |  |  |  |  |  |  |  |  |  |  |  |  | R145C | N |

Entries in red denote mutations that are present in CB2000.

### DNA metabolism

Every evolved isolate sequenced features a mutation in one or another component of the replication restart primosome (*priA*, *priC*, *dnaB*, *dnaT*), critical for the origin-independent restart of stalled DNA replication forks after recombinational DNA repair has occurred (*Cox et al., 2000*, *Marians, 2008*; *Rangarajan and Woodgate RGoodman, 2002*). A mutation in the *recA* gene appears in 13 of 22 isolates from the original four evolved populations, appearing in every evolved population except IR-3-20. The RecA D276N substitution is fixed in IR-2-20 (including CB2000), and the identical allele also appears in the separately evolved isolate from IR-4-20, CB4000. Two different variants of RecA, D276A and A289S, appear in strains derived from population IR-1-20, helping to define two of the three sub-populations exhibiting clonal interference. Mutations in the cryptic helicase gene *yfjK* are present in three of the four populations, although a mutation appears in only one of three sub-populations of IR-1-20.

### Oxidative damage suppression

Over the sequenced isolates, quite a few mutations clustered in components of the *rsxABCDGE* gene cluster and *gsiB*. The *rsx* gene cluster encodes a multi-subunit complex involved in the reduction of SoxR, part of a regulatory circuit for up to 50 target genes. Some of these are directly involved in protection against reactive oxygen species (*Demple, 1996*). The *gsiB* gene encodes an ABC transporter for glutathione. Different mutations in *gsiB* are fixed in IR-2-20 and IR-3-20.

### Cell wall biogenesis

Mutations of *wcaK* and *wcaC* appear frequently in populations IR-2-20 and IR-3-20, respectively, and also appear in *wcaA* in the single isolate from IR-4-20. The pattern is reinforced by the appearance of a *wcaL* mutation in 5 of the 9 isolates derived from the continued evolution of CB1000. The products of the *wca* operon are involved in the synthesis of secreted exopolysaccharides that confer significant resistance to heat and acid stress (*Mao et al., 2001*). Mutations in *nanE* and *nanT* are fixed in IR-2-20 and IR-3-20, respectively. The pattern is again reinforced by the appearance of a mutation in *nanA* in all nine of the sequenced isolates derived from the continued evolution of CB1000 (*Supplementary file 1B*). The *nan* operon is involved in the recycling and/or synthesis of components of the bacterial cell wall, particularly N-acetylmannosamine and N-acetylneuraminate (*Plumbridge and Vimr, 1999*).

## Defining the molecular basis of directly evolved extreme radiation resistance in CB2000

To assess the importance of these seven CB2000 mutations quantitatively, we took three approaches:

I. Mutations identified in CB2000 were moved into the radiosensitive Founder strain. This was done both individually and collectively with one or more of the other mutations. The e14 prophage was deleted (Δe14) in our wild-type background, since this genetic alteration occurs very early in the evolution of IR resistance in all of our strains, and the effects of this deletion have already been characterized (*Harris et al., 2009*). If any of the 7 mutations we identify above are contributing to the phenotype, they are doing so in a background that excludes the e14 prophage. In addition to the seven mutations identified in *Table 2*, we selected the *glpD* mutation as an additional target for analysis since it fulfilled criterion I above but did not quite fit criterion II. This mutation was fixed in population IR-2-20. Although this gene was not mutated in any other population, a mutation that might affect regulation of *glpD* was found in three isolates of IR-3-20.

II. The same eight total mutations (including *glpD*) were reverted back to the original Founder sequence in CB2000 both individually and in combination with other mutations. This allows us to determine if the mutations contribute to IR resistance in a genetic background that includes all or most of the other 69 mutations present in CB2000.

III. We deleted the nonessential genes carrying these mutations in the radiosensitive Founder strain (in a genetic background deleted for e14) individually. We wished to determine if a definitive knockout of an individual gene could mimic any observed effects of the individual mutations observed in CB2000. If the answer was yes, we reasoned that the mutation we observed in CB2000 was likely to be a loss of function mutation.

All of the strains constructed for this effort were assayed for survival to 3000 Gy to measure the contribution of individual mutations and mutation combinations to the IR resistance phenotype.

## Individual mutation contributions to the IR resistance phenotype

All of the mutations identified above made a measurable contribution to the phenotype, some large and some small. As illustrated in *Figure 2A*, moving the CB2000 alleles of *gsiB*, *rsxB*, *glpD*, and *wcaK* individually into the Founder Δe14 background resulted in small increases in radioresistance, ranging from 2 (*wcaK*)–7 (*gsiB*) fold. Each of these mutations thus appears to contribute to the IR resistance

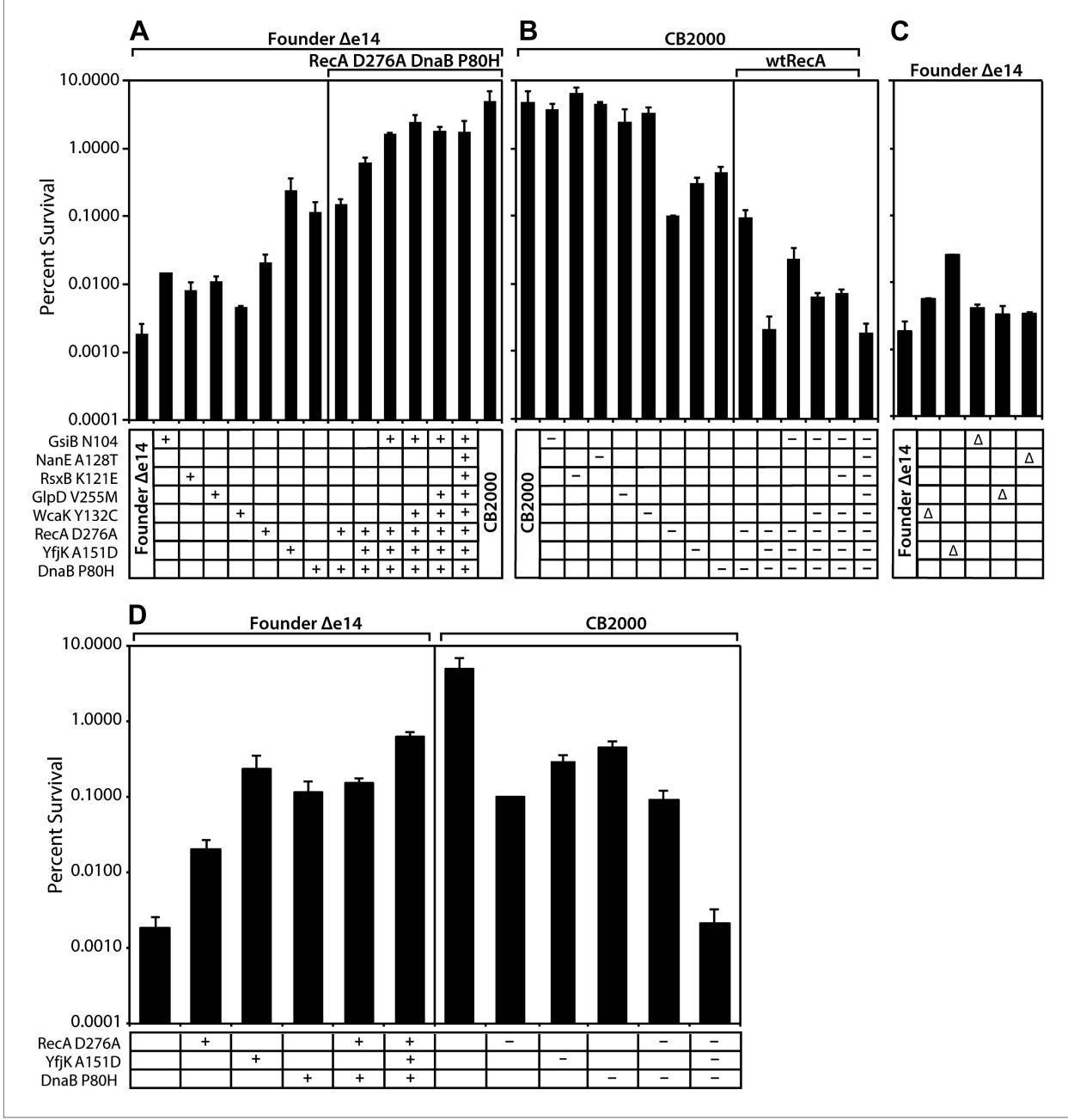

**Figure 2**. Effects of selected mutations on survival of *E. coli*. Mid-logarithmic phase cultures were irradiated to 3000 Gy and plated to measure survival as described in 'Material and methods'. (**A**) mutations discovered in CB2000 were moved individually and in combination into the Founder Δe14 background. Mutations present in a given strain are indicated by a + symbol. For reasons not understood, it proved impossible to move the *nanE* mutation into this background on its own. CB2000 itself is presented in the final lane. (**B**) the same point mutations (this time including *nanE*) were mutated back to Founder sequence in the CB2000 background. The first lane is CB2000. Mutations converted to the wild type allele in a given strain are indicated by a –symbol. (**C**) non-essential genes assayed in Panel **A** and **B** were deleted in the Founder Δe14 background, with the deleted gene indicated by a Δ symbol. **D**, The effects of mutations in genes *recA*, *dnaB*, and *yfjK* are summarized, with symbols as in panels **A**–**C**. All results were obtained with a [137]Cs irradiator (~7 Gy/min).

phenotype, but in a relatively modest way. In contrast, moving the CB2000 alleles encoding the RecA D276N, YfjK A151D, or DnaB P80H proteins individually into Founder Δe14 resulted in 1–2 log increases in IR resistance, indicating that these mutations contribute in a more substantial way to the observed IR resistance of CB2000. Of these three mutations, the *yfjK* mutation had the greatest effect when present alone in the Founder Δe14 background (*Figure 2A*; the effects of these three mutations are summarized in *Figure 2D*). The effects of the individual mutations on IR resistance in the Founder Δe14 background were echoed by the effects of reverting these same mutations to wild type in the CB2000 background (*Figure 2B*). Individually reverting the *recA*, *dnaB*, and *yfjK* alleles back to the Founder sequence resulted in 1–2 log decreases in resistance. In this case, the *recA* reversion resulted in the largest loss of resistance with an approximately 50-fold decrease in IR resistance relative to CB2000. Reverting the other mutant alleles (*gsiB, rsxB, glpD, wcaK,* and *nanE*) to wild type in the CB2000 background had significant but relatively small effects.

## Mutation combinations
Combining multiple mutations in the Founder Δe14 background allowed us to assess whether mutations were acting additively or affected single pathways. As shown in *Figure 2A*, the deletion of the e14 prophage along with point mutations in *recA*, *dnaB*, and *yfjK* accounted for much of the radiation resistance of CB2000. Of the other 5, the *gsiB* mutation appeared to contribute the most (an additional threefold) when combined with the DNA repair function mutations. A strain incorporating the *recA, dnaB, yfjK, wcaK, and gsiB* mutations accounted for all but twofold of the overall increase in radiation resistance in CB2000. Further supporting the importance of the *recA, dnaB,* and *yfjK* mutations, conversion of each of these mutations back to the wild-type allele causes a substantial loss of IR resistance (*Figure 2B*). *Figure 2B,D* illustrate that the radiation resistant phenotype observed in CB2000 was eliminated–in its entirety–by converting only these three genes back to the wild type sequence. The effects of the other mutant alleles (*gsiB, rsxB, glpD, wcaK,* and *nanE*) were small to insignificant in all cases.

Mutations in *recA* and *dnaB* appear to affect the same pathway, as a combination of the *recA* and *dnaB* mutations in the Founder *Δe14* background did not increase resistance beyond that observed for the *dnaB* mutation alone. Combining the *yfjK* mutation with those in *dnaB* and *recA* generates a further increase in IR resistance (*Figure 2A,D*), suggesting that the *yfjK* mutation affects a different cellular process or pathway than the other two. A similar pattern is seen when mutations in the CB2000 background are reverted to wild type alleles. Individual reversion of the *recA* and *dnaB* mutations in CB2000 back to the Founder sequence resulted in a ~50- and ~10-fold decrease in resistance from CB2000, respectively (*Figure 2B*). Upon reversion of the CB2000 *yfjK* mutation to wildtype, a 15-fold decrease in radiation resistance was observed (*Figure 2B*). Again, reversion of both *recA* and *dnaB* to wild type in the same cell had no more effect than one (in this case *recA*) alone. When the mutation in *yfjK* was reverted in addition to *recA* and *dnaB,* IR sensitivity increased.

Several of the key results of *Figure 2D* were confirmed in direct competition experiments (*Figure 3*) (*Lenski et al., 1991*). We incorporated a neutral Ara⁻mutation into particular strains (which confers a red color on colonies when grown on tetrazolium arabinose [TA] indicator plates) to permit color based scoring of mixed populations (*Lenski et al., 1991*). When CB2000 (Ara⁻) was mixed in a 1:1 ratio with CB2000 in which the *recA*, *dnaB*, and *yfjK* mutations had been converted to wild type, and irradiated at 2000 and 3000 Gy, CB2000 exhibited at least a 2 log advantage in survival at the higher IR dose (*Figure 3A*). Similarly, Founder Δe14 (Ara⁻) was mixed with an isogenic strain carrying the *recA*, *dnaB*, and *yfjK* mutations. A 2 log advantage for the strain incorporating the three mutations was seen after irradiation at 2000 Gy (*Figure 3B,C*). When CB2000 in which the *recA*, *dnaB*, and *yfjK* mutations had been converted to wild type was mixed with Founder Δe14 (Ara⁻), survival of the two strains was similar (the CB2000 reversion strain had less than a 0.5 log advantage [*Figure 3D,E*]).

## Mutations in wcaK, gsiB, yfjK, nanE, and glpD likely result in a loss of function
Nonessential genes mutated in these studies were individually deleted in the Founder Δe14 background and the resulting strains were assayed for survival of 3000 Gy and shown in *Figure 2C*. Strains deleted for *wcaK*, *gsiB*, *glpD*, and *nanE* respectively are twofold to threefold more resistant than Founder Δe14. For *wcaK* and *glpD*, this increase in resistance is similar to the resistance observed in Founder Δe14 strains carrying the point mutations in these genes, suggesting that the mutations result in loss of function. The twofold increase in resistance conferred by a *gsiB* deletion is less than that of the *gsiB* point mutation, suggesting a more complex situation. The situation with *yfjK* is similar. Founder

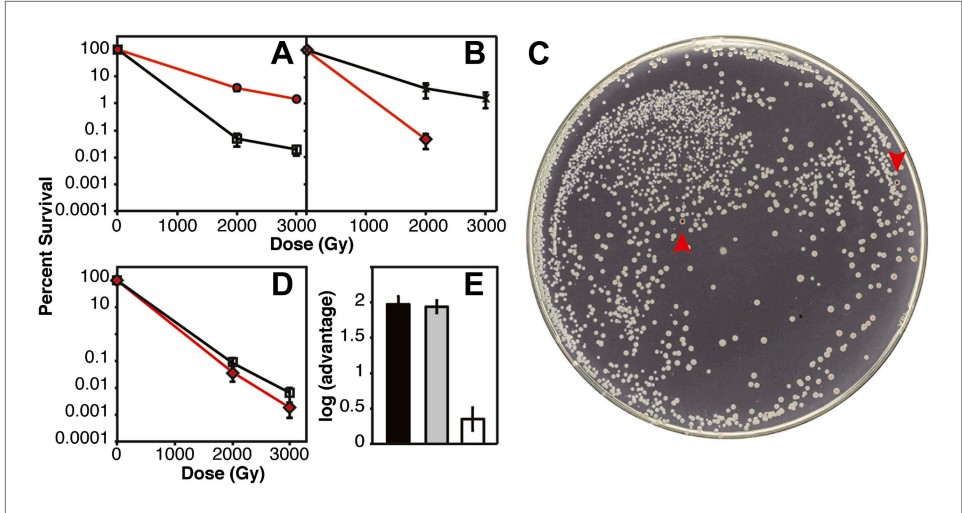

**Figure 3**. IR survival in direct competition assays. Mid-logarithmic cultures consisting of a 1:1 ratio of the competing Ara+ and Ara⁻strains were irradiated to 2000 and 3000 Gy and plated on tetrazolium arabinose indicator plates to distinguish the frequency of survival for both strains. (**A**) CB2000 Ara⁻ (red, ^), vs CB2000 wtRecA wtDnaB wtYfjK, □ (**B**) Founder Δe14 Ara⁻, (red, △), *vs* Founder Δe14 RecA D276N DnaB P80H YfjK A151D, ◇ (**C**) TA plate of 2000 Gy survival competition of Founder Δe14 Ara⁻vs Founder Δe14 RecA D276N DnaB P80H YfjK A151D. The two red Founder Δe14 Ara⁻colonies are indicated by red arrows. (**D**) Founder Δe14 Ara⁻ (red, ◇), *vs* CB2000 wtRecA wtDnaB wtYfjK, □. (**E**) Log advantage in survival to 2000 Gy of CB2000 Ara⁻over CB2000 wtRecA wtDnaB wt YfjK (in black), Founder Δe14 RecA D276N DnaB P80H YfjK P80H over Founder Δe14 Ara⁻ (in grey), and CB2000 wtRecA wtDnaB wt YfjK over Founder Δe14 Ara⁻ (in white).

Δe14 Δ*yfjK* cells are 13-fold more resistant than the isogenic parent strain as illustrated in *Figure 2C*. However, the Founder Δe14 strain encoding YfjK A151D is an additional 10-fold more resistant than the deletion, and nearly 2 orders of magnitude more resistant than Founder Δe14. It is possible that the mutation in *yfjK* is a partial loss of function, or that it eliminates one or more activities of the YfjK protein but not all of them. The effects of the *yfjK* deletion also suggests that this gene affects a process that is distinct in some manner from that affected by the *recA* and *dnaB* genes. Deletion of *recA* or *dnaB* results in slightly reduced growth or viability, or no viability, respectively, even without IR exposure.

As detailed in *Table 2*, two additional mutational patterns are evident in the four evolved populations that are not manifested in CB2000. Mutations affecting the *clpXP* complex are prominent in populations IR-1-20 and IR-3-20, and new mutations potentially affecting this system also appear in the further evolution of CB1000. *clpXP* encodes a proteolytic system involved in turnover of many key cellular proteins including many involved in DNA metabolism (*Baker and Sauer, 2006*), and is essential for the transition from growth to stationary phase. It is possible that this complex is essential for the elimination of proteins that have been inactivated by oxidative carbonylation. A mutation in *fnr* is fixed in IR-3-20. While this gene is not targeted in any of the other three original populations, its importance is suggested by the appearance of a mutation that leads to a truncated Fnr protein in all nine isolates derived from the further evolution of CB1000 (*Supplementary file 1B*). Fnr is a regulator that oversees the transition between aerobic and anaerobic metabolism (*Kiley and Beinert, 1998*).

## The IR resistance phenotype does not entail significant changes in the transcriptome, metabolome, or metal ion content

We examined the baseline metabolic profiles of the Founder strain and several of the IR resistant isolates. The strains were not subjected to irradiation prior to analysis. The results measure the state of the cells when they first encounter irradiation.

We used RNA-Seq (*Marioni et al., 2008*) to directly sequence and map RNAs that are expressed in the Founder, CB1000, and CB2000, with the comparison reported in *Supplementary file 1C*. Overall, there were few genes with different expression profiles when comparing the strains with a 1.5-fold cutoff enforced. The only commonality in expression patterns between the evolved strains was the >1.5-fold

decrease in the transcript abundance compared to Founder of the following genes: *fruBKA* (fructose metabolism), *sdaC* (serine transport) and *proK* (proline tRNA). Transcription of the entire fimbrial operon is increased in CB1000, possibly due to phase variation of the *fimS* region, but in CB2000, only the *fimC* gene exhibits an increase in transcription of more than 1.5-fold. In both evolved strains *icd* (b4519) transcript levels are increased, likely due to the excision of the e14 prophage, which reconstitutes the *icd* gene with a different 3′ end of the gene containing 2 base substitutions. In general, changes are minimal. There are few genes that display a newly constitutive level of expression that is strikingly higher than in the parent Founder strain, in spite of the existence of mutations in a number of genes encoding global regulators.

Using NMR, we investigated metabolite concentrations in the total soluble fraction of the cytosol from Founder, two evolved radioresistant strains, CB1000 and CB2000, and in CB1013 (an isolate from the IR-1-20 population that has a distinctively different mutational profile relative to CB1000) (*Figure 4A*). There are no significant changes in metabolites between any of the strains measured, with the possible exception of small apparent decreases in the levels of acetate and succinate in CB2000 as compared to Founder. Unlike *D. radiodurans*, there is no evidence of accumulation of intermediate metabolites that could act as antioxidants. The measured metabolites included glutathione, a molecule that plays a particularly important role in cellular redox chemistry.

We also used trace metal analysis to measure total metal content of all strains for which we obtained genomic sequences. Mn/Fe ratios are reported in *Figure 4B*, and complete listings of metal ion measurements are provided in *Figure 4—figure supplement 1*. In spite of the demonstrable increases in radiation resistance exhibited by all of our isolates, we did not see a significant change in the Mn/Fe ratio (nor significant increases in the concentration of either metal) in most of our directly evolved highly radioresistant strains of *E. coli*. There is a minor elevation in manganese in the one evolved isolate from population IR-4-20 (CB4000) and in some strains derived from the further evolution of CB1000. There is no universal change in metal content in the evolved strains that mirror the apparent adaptation seen in *D. radiodurans* and other IR-resistant bacteria.

## Discussion

To the three potential mechanisms described in the introduction of this article—amelioration of protein oxidation, novel DNA repair systems, and nucleoid condensation—we now document a fourth. IR resistance can be increased—dramatically—by functional enhancement and/or adaptation of existing DNA repair enzymes. Although classical DNA repair systems may be shared widely in different IR resistant and IR sensitive bacterial species, those DNA repair systems are biologically malleable. Classical DNA repair pathways can adapt to facilitate more efficient repair when cells are exposed to high levels of IR. In our evolved isolates, adaptations to DNA repair, involving genetic alterations in well-studied enzymes that are present in most bacterial species, represent a substantial and sometimes the dominant adaptation that contributes to the IR resistance phenotype. In one well-characterized isolate, three mutations—all in DNA repair functions—largely account for the increase in IR resistance.

The metabolic profile of our cells indicates that the evolved strains possess little or no unusual gene expression, metabolite concentration, or metal ion adaptations when they first encounter irradiation. We acknowledge that irradiation may lead to an alteration of the profile of genes induced in the IR resistant isolates, and those isolates may thus adapt to the challenge of irradiation more rapidly. In CB2000, changes in the *rsxB* gene might bring about some significant and beneficial changes in the IR response. However, in the case of CB2000, it is clear that changes to DNA repair genes—not regulatory genes—play the major role in the observed IR resistance.

The complement of DNA repair systems present tend to be quite similar from one bacterial species to another, but they are not identical. The differences, some subtle and some significant, inevitably reflect the lifestyle and environment that applies to each species. Whereas *Deinococcus radiodurans* has a fairly standard set of DNA repair functions (Daly, MJ, 2012), there are important distinctions in both the relevant gene catalogue (*Cox and Battista, 2005*, *Blasius et al., 2008*) and the mechanisms used to recover from exposure to high levels of ionizing radiation (*Slade et al., 2009*). In the present study, we directly demonstrate that genetic innovation involving the cellular DNA repair systems can directly contribute, and contribute substantially, to the acquisition of extreme resistance to ionizing radiation.

Nevertheless, we emphasize that there are other effects clearly evident in the data. Evolution of a complex extremophile phenotype, such as extreme resistance to ionizing radiation, has no single molecular explanation. There are multiple paths; multiple mutations affecting multiple cellular systems make significant contributions. The mutational patterns suggest that improvements in the amelioration

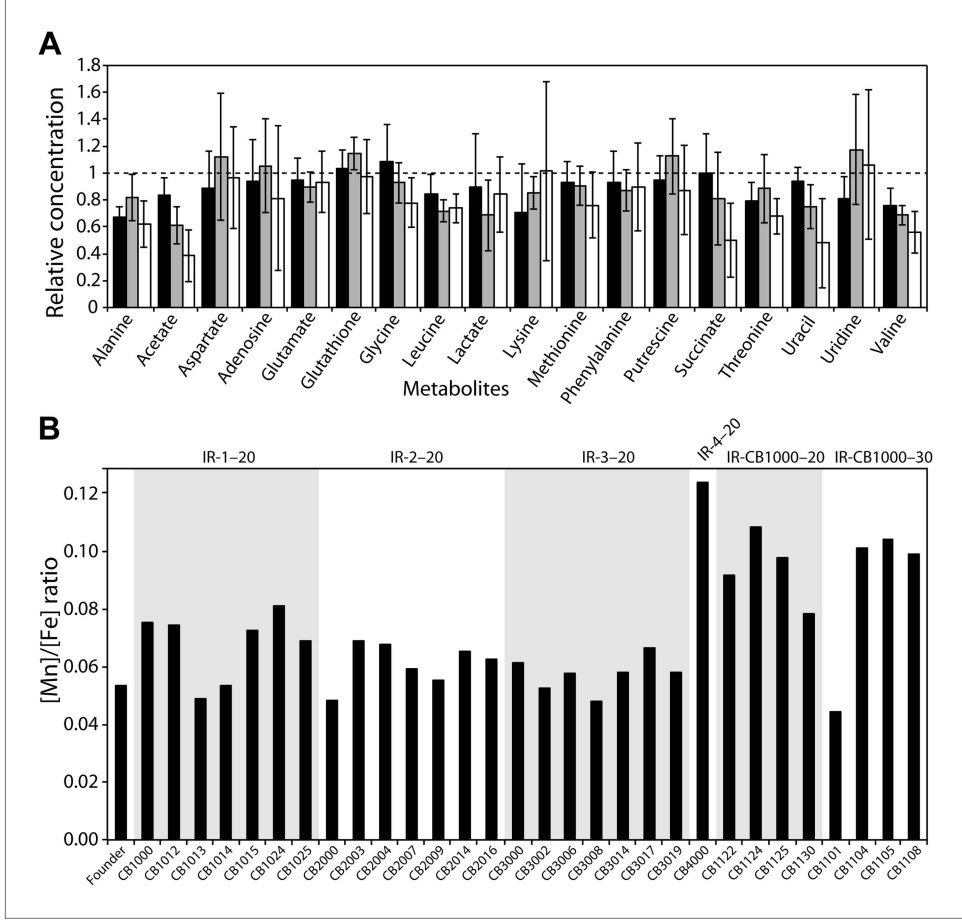

**Figure 4**. (**A**), Measurements of metabolites from three representative evolved *E. coli* strains as compared to Founder. Metabolites from whole cell pellets collected during logarithmic growth in LB were identified using a two-dimensional $^1$H-$^{13}$C Heteronuclear Single Quantum Coherence (HSQC) experiment. Each metabolite is expressed as a ratio of the amount measured in the evolved strain (CB1000, CB1013, or CB2000; black, gray, and white bars, respectively) relative to the Founder. (**B**) Ratios of manganese to iron are plotted for all isolates for which genomic sequences were obtained. The average increase in Mn/Fe ratio in strains derived from the further evolution of CB1000 is 1.4-fold.

The following figure supplements are available for figure 4:

**Figure supplement 1**. Relative levels of metals are unchanged in evolved *E. coli*.

of protein oxidation, as first pointed out by Daly et al. (***Daly et al., 2004***, ***Slade and Radman, 2011***, ***Krisko and Radman, 2010***, ***Daly, 2012***, ***2009***), may also play a role in some of the evolved populations. Although that role appears to be relatively minor in CB2000, it may well predominate in one or more of the other populations. Given the genes affected, any mechanisms contributing to the amelioration of protein oxidation in these isolates may be somewhat different than those described to date for *D. radiodurans*.

We note that the mutations to DNA repair genes might, in principle, simply render the protein products of those genes less vulnerable to oxidation. This does not appear to be the case for the changes we observe in the *recA* gene. We have characterized the proteins encoded by the *recA* gene variants described in this report (JR Piechura and MM Cox, unpublished data), particularly the RecA D276A and D276N mutant proteins. In brief, the altered RecA proteins nucleate filament formation more rapidly and extend those filaments more slowly than the wild-type protein, leading to larger numbers of shorter filaments. Those same proteins function better as shorter filaments, and are less sensitive to inhibition by ADP. In general, the proteins exhibit functional alterations that are easily rationalized in the context of a recombinational system that must simultaneously deal with large numbers of double strand breaks in cells where metabolic processes might be compromised.

Contributions to the extreme IR resistance phenotype by additional biochemical processes appear likely, and remain to be explored.

## Materials and methods

### Bacterial strains used in this study

All strains used in this study are *E. coli* K-12 derivatives and are listed in *Supplementary file 1D*. Genetic manipulations were performed by site directed mutagenesis (Stratagene) and as previously described (*Datsenko and Wanner, 2000*).

### Directed evolution of Escherichia coli

Radioresistant populations IR-1-20, IR-2-20, IR-3-20, and IR-4-20 were generated in a directed evolution experiment described previously (*Harris et al., 2009*). To further evolve the CB1000 isolate, 1 ml of mid logarithmic phase liquid culture grown in LB was placed into two 1.5 ml plastic tubes. One was archived at −80°C, and the other was exposed to IR ($^{60}$Co source from a Shepherd model 484 irradiator; 19 Gy/min). After irradiation, appropriate dilutions were plated to estimate survival. The balance of the irradiated culture was used to inoculate fresh LB broth. Survivors were grown to stationary phase (12–18 hr) and the protocol was repeated. The administered radiation dose was adjusted to allow approximately 1% survival after 1 day at 37°C, with the dose increasing as radio-resistance increased. Survivors at the end of 20 and 30 rounds of irradiation constitute a population of cells, designated IR-CB1000-20 and IR-CB1000-30, respectively. Single colony isolates from both populations were isolated and designated with the prefix 'CB'. All archived populations and strains were stored at −80°C with 15% vol/vol DMSO added as cryoprotectant.

### High-throughput sequencing using an Illumina instrument

New evolved isolates described in this study, were sequenced at the Joint Genome Institute, Walnut Creek, CA, as detailed previously (*Harris et al., 2009*). We used Illumina-based next generation sequencing that typically generates very low error levels in bacterial genome sequencing. SNP calls required their presence in at least three reads. Typically, the SNPs were based on 50–150 reads. To estimate error levels, the sequence alignments for one dataset (CB2004) were examined manually. Since this is a haploid genome, loci which contained multiple alleles (a mixture or reference and 'variant' alleles) are indications of alignment errors or sequence specific errors (*Nakamura et al., 2011*) and were not called as SNPs No false positives were identified in the manual examination and we estimate their appearance at rates of <2%. If reads can be aligned to multiple locations in the genome, their exact placement is ambiguous and assigned a map quality score of zero (MQ = 0). It is not possible to call SNPs in regions that contain only MQ = 0 reads and false negative calls are potentially present. Approximately, 100,000 bp of the genome was covered by only MQ = 0 reads, and thus the potential for false negatives extends over about 2.2% of the genome.

### IR survival assay

Cells from a fresh single colony of each strain were cultured in Luria–Bertani (LB) broth (*Miller, 1992*) at 37°C with aeration. After growth overnight, cultures were diluted 1:1000 into 10 ml fresh LB broth in 125 ml flasks and grown at 37°C with shaking until an optical density ($OD_{600}$) of ~0.2 was reached. Each culture was incubated on ice for 5 min before a 1 ml sample was transferred to an eppendorf tube and irradiated in a Mark I $^{137}$Cs irradiator (from JL Shepherd and Associates, San Fernando, CA, USA) for a time corresponding to 3 kGy (~7 Gy/min). Irradiated samples as well as the non-irradiated control samples for each culture were diluted appropriately, and plated on LB 1.5% agar medium to determine the total number of colony forming units (CFUs). Percent survival was calculated by dividing the titer of the surviving population by the titer of the non-irradiated control sample. Initial cell densities ranged from 2 to $6 \times 10^7$ CFU/ml (average $4 \times 10^7$ CFU/ml). For each strain, 3–5 biological replicates were carried out.

### IR survival competition assays

Cells from a fresh single colony of each strain were cultured in LB broth (*Miller, 1992*) at 37°C with aeration. After growth overnight, competition cultures were started by inoculating 10 ml fresh LB broth with 35 µl of competition *Ara*+ and Ara⁻ strains in 125 ml flasks and grown at 37°C with shaking until an optical density ($OD_{600}$) of ~0.2 was reached. Each competition culture was incubated on ice for 5 min before a 1 ml sample was transferred to an eppendorf tube and irradiated in a Mark I $^{137}$Cs

irradiator (from JL Shepherd and Associates, San Fernando, CA) for a time corresponding to 2 kGy and 3 kGy (~7 Gy/min). Irradiated samples were diluted appropriately, and plated on TA plates to determine the total number of surviving red and white colony forming units. A non-irradiated control sample for each competition culture was diluted and plated to determine the titer of each culture and the percent Ara+ vs Ara−cells before irradiation. For each competition, three biological replicates were carried out. Percent survival was calculated by dividing the titer of the surviving population by the titer of the non-irradiated control sample, for both *Ara+* and Ara−strains. Selection rate, r, also referred to as log(advantage), was calculated as $\log(N_{1(IR)})/(N_{1(No-IR)}) - \log(N_{2(IR)})/(N_{2(No-IR)})$, where $N_{1(No-IR)}$ and $N_{2(No-IR)}$ represent the initial densities of the two competing strains before IR treatment, and $N_{1(IR)}$ and $N_{2(IR)}$ represent the corresponding densities after IR exposure. Normally, in a direct competition experiment, plates with fewer than 20 colonies of either competitor are usually excluded to reduce the effect of outliers caused by low counts (*Breed and Dotterer, 1916*). However, because the differences in fitness after IR treatment is so great between CB2000 Ara−and CB2000 wtRecA wtDnaB wtYfjK and between Founder Δe14 Ara−and Founder Δe14 RecA D276N DnaB P80H YfjK A151D, it was virtually impossible to retrieve at least 20 colonies of the sensitive strains in a range that we could also use to calculate the density of the resistant strains. The selection rates in *Figure 3E* are approximate, because there were less than 20 colonies counted on plates for the sensitive strains. However, the trend that we show in *Figure 2* is strongly conserved. By reverting the three mutations in DNA metabolism genes, CB2000 loses virtually all of its IR resistance. We report in *Figure 3E* that when treated with 2000 Gy, CB2000 Ara− has *at least* a two log fitness advantage over CB2000 wtRecA wtDnaB wtYfjK and inversely, Founder Δe14 Ara- RecA D276N DnaB P80H YfjK P80H has at least a two log advantage over Founder Δe14. Because their sensitivities to IR were so similar, we did not have a problem with retrieving more than 20 colonies for each strain in the competition of Founder Δe14 and CB2000 wtRecA wtDnaB wtYfjK. Rather, CB2000 wtRecA wtDnaB wtYfjK had less than a half log advantage (less than threefold) over Founder Δe14, again illustrating the importance of mutations in these three genes for extreme radiation resistance.

## RNA-seq

This method is reviewed in *Croucher and Thomson (2010)*. Sample preparation: samples were prepared as described in (*Durfee et al., 2008*) with modification as described here. Cell growth: Overnight cultures from single-cell inoculates grown in LB were used to inoculate 20 ml of LB in a 125 ml flask with appropriate antibiotic to an initial $OD_{600}$ of 0.02. Cultures were grown at 37°C with aeration until an $OD_{600}$ of ~0.2 was reached. Aliquots of 15 ml of each culture were mixed with 30 ml of RNAprotect bacterial reagent (Qiagen), inverted to mix, and incubated at room temperature for 5 min. Cells were centrifuged at 4000×*g* for 20 min at 4° and the cell pellets were stored at −80°C.

## Total RNA isolation

Total RNA was isolated using the MasterPure RNA purification kit according to the manufacturer's specifications (Epicentre, Madison, WI, USA). Nucleic acid pellets were treated with 0.05 U/µl DNase I for 45 min at 37°C and then repurified with MasterPure.

## mRNA enrichment and cDNA synthesis

10 µg of total RNA was enriched for mRNA by targeted removal of rRNA using the MICROBExpress bacterial mRNA enrichment kit (Ambion). The resulting enriched mRNA was isopropanol precipitated, and the pelleted mRNA resuspended in TE. The enriched mRNA concentration was quantified by $A_{260}$ measurement on a NanoDrop 1000 instrument. 10 µg of purified total RNA was reverse transcribed using the Superscript II double-stranded cDNA kit (Invitrogen) followed by RNase digestion and cDNA purification by phenol chloroform extraction and precipitation.

## Illumina sequencing and analysis

cDNA samples were submitted to JGI for library preparation and sequencing using the Illumina Genome Analyzer IIx to generate single-ended 36 bp reads. Libraries were prepared for sequencing according to the manufacturer's instructions.

## Analysis

Analysis was performed using the CLC-Bio Genomics Workbench version 3.7. There were two biological replicates for each of the three samples (Founder, CB1000, CB2000). Read ends were

trimmed to remove low quality and ambiguous bases and all reads less than 20 nt were discarded. Trimmed reads were mapped to the annotated CDSs of the reference genome (*E. coli* K-12 MG1655 m56 reference genome, RefSeq Accession Number NC_000913.2) with two mismatches allowed and 10 bases of each read were allowed to map beyond ORF boundaries. Expression was calculated independently for each duplicate sample. Any read that could be mapped to more than four locations was discarded. Genes encoding rRNA and tRNA transcripts were masked by removing their annotations from the reference genome prior to mapping so that did not affect normalization expression estimates of protein-coding genes. Expression values were reported in RPKM (*Mortazavi et al., 2008*).

## Differential expression

Differential expression, reported as fold-change, was determined by separately comparing the expression estimates of each pair of duplicate samples to the control samples. CB1000 and CB2000 were also directly compared. Prior to analysis, all expression values (E) were transformed into $\log_2(E+1)$ and then standardized by adjusting each sample to the expression level corresponding to 1 million mapped reads. Fold changes were tested using Baggerly's test (*Baggerly et al., 2003*) on the fold changes estimated both from the original RPKM values and the transformed/standardized values. The resulting p-values were independently corrected using the Bonferroni method and via a determination of the false discovery rate (FDR). Heat maps were generated from the transformed/standardized expression values and dendrograms showing the clustering of genes were computed using complete linkage with Pearson product–moment correlations as the distance metric.

## Growth of *E. coli* for metal analysis

Flasks were soaked in 2 N nitric acid for 12 hr and then transferred into 1% (vol/vol) nitric acid for 24 hr and rinsed in high-purity water to minimize metal contamination. Cells of CB1000, CB2000, CB3000, and CB4000 were cultured in LB broth at 37°C (*Miller, 1992*) with aeration in polypropylene tubes. After growth overnight, cultures were diluted 1:100 into 25 ml fresh LB broth in 50 ml polypropylene tubes and grown at 37°C with shaking until an optical density ($OD_{600}$) of ~0.2 was reached. This was performed in triplicate for each strain. Cultures were chilled for 10 min on ice before 10 ml were pelleted and resuspended in 1 ml fresh LB. Aliquots of 0.6 ml of cells were transferred to treated microcentrifuge tubes prepared by the metal analysis facility and pelleted. Cell pellets were stored at −20°C. To determine the titer of each cell pellet, appropriate dilutions were plated on LB 15% agar medium and incubated overnight at 37°C.

## Metal analysis

Cell pellets were submitted for analysis in acid-cleaned polypropylene vials and treated at 40°C with ultrapure $HNO_3$ and subsequently diluted to volume for analysis with 2% $HNO_3$. Samples were analyzed in the Trace Element Clean Laboratory at the Wisconsin State Laboratory of Hygiene, Madison, WI, using high-resolution inductively coupled plasma mass spectrometry. Each sample was measured twice. The metal content is reported as µg per pellet of bacteria submitted.

## NMR sample preparation

Overnight cultures of Founder, CB1000, CB2000, and CB1013 were diluted into 1L M9 media containing glucose in 6-L flasks to an initial $OD_{600}$ of 0.05 and grown at 37°C with aeration. When an $OD_{600}$ of ~0.80 was achieved, cultures were chilled on ice for 30 min before being centrifuged at 8000 rpm for 10 min at 4°C in a JLA 8.1 rotor. The supernatant was discarded, and cell pellets were washed in 25 ml of 1X M9 salts and transferred to a JA-20 tube before centrifuged 20 min at 5000×*g* at 4°C. The supernatant was discarded and 16 ml of boiling water with 250 µM MES (2-(N-morpholino)ethanesulfonic acid) was added to each pellet, vortexed briefly to loosen the pellet, and placed in boiling water for 7.5 min. Tubes were briefly vortexed again, and then centrifuged in a JA-20 rotor at 7000×*g* for 20 min at 4°C to clear cell debris. The supernatant was poured off into a clean 50-ml sterile polypropylene tube and frozen.

## NMR sample processing

Supernatants were transferred to microfilters (Sartorius Stedim Vivaspin 20, 3000 MWCO). The low MW fraction was frozen and lyophilized. Dried metabolites were dissolved in 800 µl $D_2O$ containing 300 µl DSS and 300 µl $NaN_3$ and titrated to pH 7.40 (±0.01) with DCl/NaOD as needed. Samples were

transferred to 5-mm NMR tubes (Wilmad Lab Glass, Vineland, NJ, USA). Spectroscopy was performed at the Nuclear Magnetic Resonance Facility at Madison (NMRFAM) in Madison, WI on a 600 MHz Varian Spectrometer with a cryoprobe and VNMRJ software. Two-dimensional $^1$H-$^{13}$C HSQC spectra were acquired using 4 transits, 32 steady state transits, 0.3 s acquisition time, and 512 increments.

## NMR analysis

Analysis was performed using rNMR, an open source software package developed at NMRFAM (*Lewis et al., 2009*). To quantify signals, standard compounds of the observed metabolites were prepared at 2, 5, and 10 mM. The resonances from these compounds were linearly regressed in order to measure concentration as a function of intensity. The samples were normalized to 5 mM MES and their resultant peak intensities used to obtain concentrations of the measured metabolites using the regression slopes.

# Additional information

### Funding

| Funder | Grant reference number | Author |
|---|---|---|
| National Institutes of Health | GM32335 | Michael M Cox |
| Department of Energy | DEFG0201ER63151, CSP2009.796601 | John R Battista |

The funders had no role in study design, data collection and interpretation, or the decision to submit the work for publication.

### Author contributions

RTB, AJK, Conception and design, Acquisition of data, Analysis and interpretation of data, Drafting or revising the article; ELC, NTP, Analysis and interpretation of data, Drafting or revising the article; WSS, CP, LAP, Acquisition of data, Analysis and interpretation of data, Drafting or revising the article; JAM, JM, ZW, EAW, Acquisition of data, Analysis and interpretation of data; JRB, MMC, Conception and design, Analysis and interpretation of data, Drafting or revising the article

# Additional files

### Supplementary files

• Supplementary file 1. (**A**) Mutation table for isolates sequenced from IR-1-20, IR-2-20, IR-3-20, and IR-4-20. (**B**) Mutation table for isolates sequenced from IR-CB1000-20 and IR-CB1000-30. (**C**) RNA-SEQ results for CB1000 and CB2000 as compared to Founder. (**D**) Strains used in this study.

### Major dataset

The following dataset was generated:

| Author(s) | Year | Dataset title | Dataset ID and/or URL | Database, license, and accessibility information |
|---|---|---|---|---|
| Blattner FR, et al. | 2004 | E. coli K-12 MG1655 m56 reference genome | NC_00,913.2; http://www.ncbi.nlm.nih.gov/assembly/GCF_000005845.1/ | Publicly available at the NCBI Assembly database (http://www.ncbi.nlm.nih.gov/assembly/). |

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
