## [Decision Letter]

Thank you for sending your work entitled “Evolution of Extreme Resistance to Ionizing Radiation via Genetic Adaptation of DNA Repair” for consideration at *eLife*. Your article has been favorably evaluated by a Senior editor, Detlef Weigel, and 3 reviewers.

The editor and reviewers discussed their comments before we reached this decision, and the Reviewing editor has assembled the following comments to help you prepare a revised submission.

This paper extends previous findings from the work of [18], which described *E. coli* populations with unusually high tolerance to ionizing radiation (IR) generated through directional selection in the laboratory. Using genetic analyses, the authors show that three out of the 69 mutations found in one lineage account for most of the IR tolerance on top of the one found in the starting strain (with an e14 prophage deletion). All three mutations are involved in DNA repair and replication. The authors convincingly conclude that genetic innovations involving pre-existing DNA repair functions likely play a large role in the acquisition of unusually high IR tolerance.

Interestingly, the *recA dnaB* seems to be not more IR tolerant than the *dnaB* single mutant, although it is more resistant than the *recA* single mutant. This part of the work would, however, be more convincing if ALL possible single and double mutations in the founder and ALL possible and single reversions in the CB2000 strain were examined, completing the matrix shown in Figure 1.

Additional major concerns that need to addressed during revision include following:

1) What are the false negative/false positive rates for mutation detection?

2) Throughout the manuscript, survival is always reported as fraction of the initial cell density. However, no mention is made of whether or not cell yields change over time. Do all mutant strains under all conditions always start at the same cell density? Someplace in the paper it is necessary to report actual CFU/ml measurements. Two strains might both experience 1% survival, but this is perhaps less meaningful it they start out at very different population densities.

3) Related to comment #1, there is another concern, based on the authors’ comments about population CB1000-30, which “continued” to grow during irradiation. The normal protocol is to grow cells to mid-log, then irradiate them in their culture medium, presumably at room temperature, and then sample. However, this one isolate continues to grow during this protocol. Does this mutant grow better under all conditions tested, e.g., at room temperature vs 37°C? Is there some link between this strains altered growth ability and its IR resistance?

4) The authors discuss the complex population structures of the evolved populations. Can they comment on the difference in IR resistance between members of the same population? Just how variable are these populations in general? The reviewers realize that only a subset of strains get sequenced, but they are curious as to the overall diversity of radio-resistance phenotypes.

5) In their discussion of the “genetic basis” of IR resistance, the authors focus on the role of repair mechanisms, among others, in their evolved strains. Given that all of the strains sequenced have many more mutations than those shown to be causal for the resistance phenotype, I wonder if the basal mutation frequency of the evolved strains has changed with respect to the founder in the absence of IR? For example, is the spontaneous mutation frequency to rifampicin resistance the same in the founder vs evolved strains? Does this affect the “evolvability” of these strains under positive selection during the IR passage protocol?

---

## [Author Response]

*Interestingly, the* recA dnaB *seems to be not more IR tolerant than the* dnaB *single mutant, although it is more resistant than the* recA *single mutant. This part of the work would, however, be more convincing if ALL possible single and double mutations in the founder and ALL possible and single reversions in the CB2000 strain were examined, completing the matrix shown in*
Figure 1*.*

We presume that this comment is directed at our suggestion that *yfjK* might affect a different pathway or process than *recA* or *dnaB*. We have added the single mutants of *yfjK* to Figure 1. Due to personnel availability and technical issues, construction of the various double mutants that might include *yfjK* and testing them would require several additional months. Since these are interesting but would not address any central thesis of the paper, we have elected to move ahead without them. We note that the *yfjK* story is complex, and will be one subject of a study soon to get underway. It does seem like *yfjK* is distinct from *recA* and *dnaB*. A *yfjK* deletion (Figure 1) results in a significant increase in IR resistance, whereas *recA* deletions exhibit slightly reduced growth and viability even without IR exposure. The *dnaB* gene is essential and cannot be deleted. This point is now made in the revised text.

We note that the original Figure 1 is now 2D due to the inclusion of a new Figure 1 and renumbering of the remaining figures in response to a different referee comment later in this sequence.

*1) What are the false negative/false positive rates for mutation detection*?

When using Illumina-based next generation sequencing with clean data (JGI) and a very high rate of coverage, the error rate for bacterial genomics work should be low. Although generating an estimate of false positives and false negatives is not trivial, we tried to address this in the Materials and methods section, in a new section.

*2) Throughout the manuscript, survival is always reported as fraction of the initial cell density. However, no mention is made of whether or not cell yields change over time. Do all mutant strains under all conditions always start at the same cell density? Someplace in the paper it is necessary to report actual CFU/ml measurements. Two strains might both experience 1% survival, but this is perhaps less meaningful it they start out at very different population densities*.

The CFU/mL for the cultures prior to irradiation were fairly consistent, as now detailed (with CFUs reported) near the end of the IR survival section of the Materials and methods section. The % survival was always measured relative to this measured starting level. IR-inflicted damage is a stochastic process. Small changes in the starting cell density should not affect the kinds and levels of damage inflicted as long as the cells are in log phase and thus in a similar metabolic state. We have previously shown (Harris 2009) that each of the four evolved populations highlighted in this study (IR-1-20, IR-2-20, IR-3-20, and IR-4-20) grow at the same rate, presumably because the outgrowth phase of each selection cycle would tend to select against any slow growers.

*3) Related to comment #1, there is another concern, based on the authors’ comments about population CB1000-30, which “continued” to grow during irradiation. The normal protocol is to grow cells to mid-log, then irradiate them in their culture medium, presumably at room temperature, and then sample. However, this one isolate continues to grow during this protocol. Does this mutant grow better under all conditions tested, e.g., at room temperature vs 37°C? Is there some link between this strains altered growth ability and its IR resistance*?

These are extremely interesting questions that, unfortunately, we cannot answer at the moment. We note that CB1000-30 is not an isolate, but a rather complex population (derived from CB1000) that has been subjected to 30 additional cycles of irradiation-driven selection. The cells in this population have 3-4 times as many mutations as do the cells in the populations we focus on the most in this paper (as stated in the 2^nd^ paragraph of the Results). We suspect that this population may either repair its DNA faster or may have lost some mechanism that normally constrains growth after damage, but a great deal of work would be needed to establish these or some other mechanism. The levels of radiation resistance in this more evolved population mean that delivering sufficient IR to kill 90% of the cells requires irradiation for many hours using the ^60^Co source currently available. A higher dose rate may well constrain growth. We do not have the capacity to control temperature in the irradiator.

The evolved population CB1000-30 (one might call it “hyper-evolved” relative to the main evolved populations IR-1-20, IR-2-20, IR-3-20, and IR-4-20, although we have not used that term) has not been examined in detail, and is used here mainly to reinforce patterns seen in the main four evolved populations that we focus on in this paper. CB1000-30 actually grows slower than the Founder strain under all conditions when radiation is not present. Given that the outgrowth stages of each selection cycle would tend to select against slow growers, we think this slower growth is significant and may be associated with another genetic innovation that makes a substantial contribution to fitness with respect to IR resistance in these cells. This is an issue that we will be examining in ongoing work.

*4) The authors discuss the complex population structures of the evolved populations. Can they comment on the difference in IR resistance between members of the same population? Just how variable are these populations in general? The reviewers realize that only a subset of strains get sequenced, but they are curious as to the overall diversity of radio-resistance phenotypes*.

There is indeed some variability. We reported on this to an extent in [18]. There, we examined survival of 62 separate isolates of population IR-1-20 at 5,000 Gy, and found about a 5 fold variation in survival among these. This is now mentioned at the end of the penultimate paragraph of the Introduction.

*5) In their discussion of the “genetic basis” of IR resistance, the authors focus on the role of repair mechanisms, among others, in their evolved strains. Given that all of the strains sequenced have many more mutations than those shown to be causal for the resistance phenotype, I wonder if the basal mutation frequency of the evolved strains has changed with respect to the founder in the absence of IR? For example, is the spontaneous mutation frequency to rifampicin resistance the same in the founder vs evolved strains? Does this affect the “evolvability” of these strains under positive selection during the IR passage protocol*?

In Harris et al., we reported that the strains examined in that study (including about a third of the isolates that are utilized in the current work) are not mutators as measured by a Rif reversion assay, and have no elevated level of mutation generation relative to the Founder. It is possible of course that mutators appear and flourish in the different populations for short times during the selection process. This is another question we are interested in exploring much more deeply in continuing work.